# Symptom Occurrence and Distress after Heart Transplantation—A Nationwide Cross-Sectional Cohort Study

**DOI:** 10.3390/ijerph17218052

**Published:** 2020-11-01

**Authors:** Marita Dalvindt, Shahab Nozohoor, Annika Kisch, Annette Lennerling, Anna Forsberg

**Affiliations:** 1Department of Cardiothoracic Surgery, Skåne University Hospital, 222 42 Lund, Sweden; Shahab.nozohoor@med.lu.se (S.N.); anna.forsberg@med.lu.se (A.F.); 2Care in High Tech Environments, Institute of Health Sciences at Lund University, 221 00 Lund, Sweden; annika.kisch@skane.se; 3Department of Haematology, Skåne University, 222 42 Lund, Sweden; 4The Transplant Centre, Sahlgrenska University Hospital, 413 45 Gothenburg, Sweden; annette.lennerling@vgregion.se; 5Institute of Health and Care Sciences, The Sahlgrenska Academy, University of Gothenburg, 405 30 Gothenburg, Sweden

**Keywords:** heart transplantation, heart recipient, symptom occurrence, symptom distress, psychological well-being, transplant specific well-being, self-management, symptom management

## Abstract

Experiencing symptoms after heart transplantation may hamper the heart recipient’s self-management which can lead to negative effects. We know little about symptom occurrence and distress after heart transplantation, especially in relation to sociodemographic variables. The aim of the study was to explore self-reported symptom occurrence and distress after heart transplantation and their relationship with self-reported psychological well-being and sociodemographic factors. This multicenter, cross-sectional, cohort study is associated with the Swedish national Self-Management After Thoracic Transplantation study (SMATT). Two questionnaires were distributed at the heart recipients’ yearly follow-up, one to five years post-transplant at three Swedish university hospitals from 2014–2017. In a total 79 heart recipients, 54 men and 25 women, with a mean age 53 years returned the questionnaires. Symptoms occurred differently depending on type and duration of follow-up. The most common symptoms, trembling hands, and decreased libido were also the most distressing. Heart recipients most burdened by symptoms were those younger than 50 years, not working, with poor psychological well-being or living alone. Fatigue explained more than 60% of the variation in transplant specific well-being. In conclusion this study points at the target groups within the heart transplant population that needs person centered symptom management support where the focus should be on side-effects of the medication i.e., trembling hands as well as the patients’ sexual health.

## 1. Introduction

The rationale behind this study is the need to focus on the illness experience by means of the relationship between symptom distress, psychological well-being and relevant sociodemographic variables rather than on the side-effects of immunosuppressive medication, which is an inevitable part of transplant medicine. Heart transplantation (HTx) is a well-established treatment for patients with end-stage heart failure, where the overall goal is prolonged survival as well as improved Quality of Life (QoL) [1]. The median survival in adult heart recipients globally is 10.7 years [2]. However, recovery after HTx is challenging and many heart recipients (HTRs) struggle with various surgery related symptoms, side-effects of the lifelong immunosuppressive medication [3], as well as the mental challenges inherent in the existential situation of being a HTR [4,5]. An assessment of symptom occurrence and distress in relation to relevant sociodemographic variables and psychological well-being presumably enables target-oriented health promotion by transplant professionals and enhances the possibility of identifying those who require tailored and person-centred self-management support.

Symptoms such as pain, depression, gastrointestinal symptoms, poor oral health, and sexual dysfunction have been found to affect the QoL of HTRs [6]. Depression among HTRs is common and increases the risk of mortality [7], while chronic pain affects as many as 75% of HTRs [8]. The view that individuals are responsible for their own health gained greater interest in the latter half of the 20th century [9] and has also been adopted within transplant medicine and nursing [10,11]. It involves a shift where organ transplantation is viewed as a chronic condition [12] demanding extensive self-management.

Self-management is defined as the ability of the individual, supported by family members, the community and healthcare professionals, to manage symptoms, treatment, lifestyle changes and the psychosocial, cultural, and spiritual consequences of chronic diseases [12]. Self-management has been adopted by transplant professionals as a framework for efficient support to transplant recipients in managing their chronic condition, namely the transplantation [10,11]. There is an expectation that the HTR should manage a multitude of behavioral and occupational changes in everyday life. However, the unfamiliar health and life situation of being an HTR has been described as a source of uncertainty and possibly of distress [4]. Furthermore, it has also been argued that an extensive symptom burden might reduce performance, especially physical performance, thus constituting a barrier to self-efficacy and subsequently self-management [13]. A vital part of self-management is symptom management, which is the process that individuals, e.g., HTRs, use in conscious attempts to gain control over their symptom burden rather than being controlled by it [14]. Symptoms are subjective experiences reflecting changes in an individual’s biopsychosocial functioning, sensations, or cognition [15]. Self-reported ratings are the preferred method for measuring symptoms [15,16]. Poorly controlled symptoms can have a devastating and costly impact on patients and healthcare systems [9]. Therefore, symptom distress is also a target area for investigations and interventions after HTx. Furthermore, it is suggested that further research on symptom distress after HTx should focus on subgroups to better understand symptom occurrence and the phenomenon of distress [3].

Therefore, the aim of this study was to explore self-reported symptom occurrence and distress after heart transplantation and their relationship with self-reported psychological well-being and sociodemographic factors.

## 2. Patients and Methods

This study was part of the Swedish national multicentre ”Self-management after thoracic transplantation” study (SMATT). The design was cross-sectional with the following inclusion criteria; being an HTR transplanted at either of the two thoracic transplant centres in Sweden performing HTx, Swedish speaking, not hospitalized or being treated for on-going acute rejection and mentally lucid. The reasons for exclusion were language barrier, being transplanted with more than one solid organ or poor health status such as being diagnosed with cancer.

### 2.1. Data Collection

Data were collected prospectively from 2014–2017 at the HTR’s yearly follow-up occurring 1–5 years post-transplant. Out of a total of 303 HTRs who had undergone transplantation during the previous five years, 153 were invited to participate and 90 (58%) were included in the study. Apart from not fulfilling the inclusion criteria, those not invited were mainly followed-up at other cardiac outpatient clinics in Sweden. The questionnaires were handed out by nurses and an occupational therapist at the transplant out-patient clinic at the HTRs’ yearly follow-up at one of the three Swedish university hospitals that perform most of such follow-ups, i.e., the Skåne University Hospital, the Sahlgrenska University Hospital and the Karolinska University Hospital. Ten HTRs did not return their questionnaires. No reminder was sent because of the high turnover among nurses at the follow-up clinics during the data collection period. The final study group consisted of 79 HTRs followed for 1 year (*n* = 28), 2 years (*n* = 17), 3 years (*n* = 11), 4 years (*n* = 17) or 5 years (*n* = 6). Their mean age was 52.68 years (SD 14.63) (range 19–72). The gender distribution which reflect the gender ratio within the Swedish heart recipient population and clinical characteristics are presented in Table 1.

#### Instruments

Two different instruments were used in the study:To explore psychological well-being and illness the Swedish version of the Psychological General Well-Being (PGWB) instrument was used [17]. This instrument consists of 22 items constituting six dimensions: anxiety, depressed mood, general health, positive well-being, self-control, and vitality. It contains a six-point scale with each domain comprising three to five items. The timeframe is specified as the previous seven days. A normal PGWB sum score is between 100 and 105 and the highest sum score is 132 [18]. Poor psychological well-being is identified if the sum score is below 100 [19]. The instrument has good internal consistency, where Cronbach’s Alpha ranges from 0.61–0.89. [17].To explore transplant specific well-being and symptom distress, The Organ Transplant Symptom and Well-being Instrument (OTSWI) was used [20]. The instrument comprises 20 questions constituting eight factors that measure fatigue, joint and muscle pain, cognitive functioning, basic activities of daily life (BADL), sleep problems, mood, foot pain and financial situation. The internal convergent validity was satisfactory, the item-scale discriminatory validity was good and together the eight factors accounted for 86% of the variance. Each response relates to the discomfort of a situation or problem, assessed on a five-point scale ranging from “not at all (0), “a little” (1), “somewhat” (2) and “quite a bit” (3) to “very much” (4). The timeframe is specified as the previous seven days. The scale has a sum score of 0–80 where lower scores indicate higher well-being. The OTSWI also measures symptom distress by the degree of discomfort from twenty transplant specific symptoms graded in the same way as the previous scale from “not at all” (0), to “very much” (4) [20].

### 2.2. Statistical Analysis

The SPSS Statistics 23 (SPSS Inc., IBM Corporation, Armonk, NY, USA) was used for analysing data. Single scale ordered category data were summarized with medians and percentiles (P_25_, P_75_). When applicable, values of *p* < 0.05 (two-tailed) were considered statistically significant. The steps in the analysis were as follows:Explore proportions (Chi square test) and describe the occurrence of symptoms and symptom distress.Explore possible differences between two unpaired groups, e.g., men and women, by means of the Mann Whitney U test.(1)Explore possible relationships by means of the Spearman’s rho test.(2)Linear multiple regression was used to assess dimensions that could explain the variation in transplant related well-being (OTSWI-sum) without controlling for the influence of age and gender as there were no gender or age differences in the dimensions assessed.(3)Logistic regression was adopted specifically to assess how much of the variance in Psychological General Well-Being (PGWB-sum) was explained by sleep problems and fatigue, where the PGWB-sum constituted the categorical dependent variable.

Age was dichotomised into two groups, younger or older than 50 years, based on the mean age.

Time on ventilator was divided in two groups, more or less than 48 h, which is a clinically established cut off point. PGWB-sum was dichotomized into poor or good well-being with a cut-off point above or below a sum score of 105 in accordance with the established cut off for the instrument.

### 2.3. Ethical Considerations

The study was approved by the Regional Ethics Board (2014/124) with supplementary approval from the Swedish Ethical Review Authority (2019/02769). The participants gave their written informed consent and the information they provided was kept confidentially and stored by the researchers in accordance with the Swedish personal data act. The participants were informed that they could withdraw at any time without consequences for their further care.

## 3. Results

Overall symptom occurrence and symptom distress is presented in Table 2.

From the first part of OTSWI covering eight transplant specific dimensions the following findings was revealed. In total, 86% reported some form of sleep problems, 73% fatigue, 65% joint and muscle pain, 63% impaired cognitive functioning, 51% some level of worry about their financial situation, 46% foot pain, 42% mood problems and 12% reduced basic ADL functioning. Table 3 presents the symptom occurrence separately for each follow-up year.

The most prevalent symptoms were trembling hands, decreased libido and being breathless with some overlap between the three most distressing symptoms as shown in Table 4.

### 3.1. Symptom Occurrence in the Different Sub-Groups

Details of the socio-demographic data are presented in Table 5.

Overall, the younger group (18–49 years) reported more symptom distress than the older group (50 years and older). The younger HTRs reported more concerns regarding their financial situation than the older HTRs (*p* ≤ 0.001). The younger group also reported being nauseous to a greater extent (*p* = 0.028), as well as more problems with diarrhoea (*p* = 0.032), headache (*p* = 0.009), trembling hands (*p* = 0.036) and feeling sad (*p* = 0.025). In addition, they felt embarrassed by their looks to a greater extent than the older HTRs (*p* = 0.014). The older group reported more distress due to decreased libido (*p* = 0.023) than the younger HTRs.

HTRs who were not working (n = 23) reported significantly more fatigue than those working part or full time (n=55) (*p* = 0.002) as well as more problems with cognitive functioning (*p* = 0.035). Those not working also reported lower BADL (*p* ≤ 0.001) and worse transplant specific well-being (*p* = 0.003) as demonstrated by a higher OTSWI sum-score compared to those who had returned to work.

The HTRs who were single or living alone reported more fatigue than those co-habiting (*p* = 0.037). They also described more problems with being breathless (*p* = 0.001) and forced to rest because of breathlessness (*p* = 0.035). Furthermore, the HTRs living alone reported more problems with dyspepsia (*p* = 0.043), headache (*p* = 0.047), numbness and stabbing pain in hands (*p* = 0.006), trembling hands (*p* = 0.001) and feeling sad (*p* = 0.029).

Regarding educational levels, there were no differences between the HTRs with higher versus compulsory education.

### 3.2. Gender Differences

Women were slightly more embarrassed by their looks than men (*p* = 0.033) and suffered marginally more from nausea (*p* = 0.003). Men reported slightly more problems than women with increased appetite (*p* = 0.047). No other gender differences were identified.

### 3.3. Clinical Factors

The HTRs treated with a Ventricular Assist Device (VAD) prior to transplantation reported more problems with fatigue (*p* = 0.019) as well as decreased appetite (*p* = 0.019) compared to those without VAD. Those on a ventilator >48 h after the HTx reported more problems with decreased libido (*p* = 0.012), while those on a ventilator <48 h reported more breathlessness (*p* = 0.064).

### 3.4. Symptom Burden and Psychological Well-Being

Those reporting good psychological well-being also had better transplant specific well-being, i.e., a lower OTSWI sum-score (*p* = 0.001), compared to those with poor psychological well-being as presented in Table 6.

### 3.5. Relationships and Possible Predictors

There was a strong relationship between sleep problems and the OTSWI-sum (r_s_ = 0.613), fatigue and the OTSWI-sum (r_s_ = 0.824), vitality and the OTSWI-sum (r_s_ = 0.707) and the PGWB-sum and the OTSWI-sum (r_s_ = −0.692).

Multiple linear regression was used to assess the extent to which sleep problems, fatigue and vitality explained the variance in transplant specific well-being (OTSWI-sum). As there were no gender or age differences in the independent variables we decided not to control for age and gender. Preliminary analyses were conducted to ensure no violation of the assumptions of normality, linearity, multicollinearity, and homoscedasticity. When only entering sleep problems in the regression model it explained 41.1% of the variance in the OTSWI-sum. Proceeding with a new model based on the above strong correlations, we entered fatigue, vitality, and the PGWB-sum to assess which variable contributed most to explaining the variance in the OTSWI-sum. This model explained 61.8% of the variance in the OTSWI-sum, *F* (3,71) = 40.96, *p* = ≤0.001. However, only one of the control measures was statistically significant, with fatigue recording the highest beta value (*beta* = 0.717, *p* = ≤0.001).

Direct logistic regression was performed to assess the impact of several factors on the likelihood that HTRs would report poor psychological wellbeing. The model contained three independent variables (gender, age, and fatigue). The full model containing all predictors was statistically significant *X*^2^ (3, N = 76) = 19.487, *p* < 0.0005, indicating that it was able to distinguish between respondents who reported or did not report psychological well-being. The model explained between 22.6% (Cox and Snell R Square) and 30.2% (Nagelkerke R Squared) of the variance in psychological well-being and correctly classified 71.1% of cases. As shown in Table 7 only two of the independent variables made a unique statistically significant contribution to the model (age and fatigue). The strongest predictor of reporting poor psychological well-being after HTx was fatigue, with an odds ratio of 1.43. This indicated that HTRs who experienced fatigue were over 1.43 times more likely to report poor psychological well-being than those who did not report fatigue when controlling for all other factors in the model, i.e., gender and age, as shown in Table 7.

## 4. Discussion

The main findings in this study were:Symptom occurrence after HTx varies depending on type of symptom and follow-up year. Trembling hands and decreased libido are prominent regardless of follow-up time, while other symptoms are more common in the first year after transplantation, i.e., feeling breathless or bloated.The most common symptoms, trembling hands and decreased libido are also the most distressing.The HTRs most burdened by symptoms are most likely to be found among those younger than 50 years, who are not working, have poor psychological well-being or live alone.Fatigue explains more than 60% of the variation in transplant specific well-being (OTSWI-sum) followed by sleep problems.

Symptom burden after solid organ transplantation [3] and HTx has been reported for some time and the results in this study add a road map of what to expect at each yearly follow-up. They also indicate that the symptoms causing the most distress are also the most prevalent and some sub-groups have a more prominent symptom burden, i.e., younger HTRs, those who are single, not working or have poor psychological well-being.

### 4.1. The Magnitude of the Problem

In general, symptom occurrence is limited after HTx, but the impact differs depending on follow-up year. As expected, the symptom distress is highest in the first year after HTx, which calls for actions from the transplant professionals who should be aware of the situation due to the extensive follow-up protocol. It is well known that gastrointestinal side-effects of the Mycophenolate Mofetil immune suppressor is common [21], which is reflected in our results. Depression and anxiety are also common problems after solid organ transplantation and increase the risk of morbidity and mortality [7]. Sleep problems, fatigue, muscle and joint pain, dyspnoea, trembling hands, increased appetite, and decreased libido were reported as distressing symptoms among solid organ recipients [20], which is confirmed by our findings. Also, Chou et al., [22] have highlighted the consequences of fatigue after HTx, showing that fatigue interference had a greater influence on QoL domains than the actual fatigue intensity. Thus, the degree to which fatigue interferes with daily life should also be assessed when exploring fatigue in this patient group.

### 4.2. Symptom Occurrence in Different Sub-Groups

The HTRs most burdened are those younger than 50 years, not working and living alone. This pattern was also seen regarding chronic pain after HTx [8]. Could it be that younger HTRs experience a pressure to return to work as well as having different conditions regarding family life and parental strain, leading to a greater challenge in terms of social adaptation? In contrast to previous research [3], there were no differences regarding educational level. Moreover, the HTRs reporting decreased libido were older than 50 years in the present study, whilst younger HTRs reported erectile dysfunction in earlier research [3]. In this study libido is understood as the person’s sexual desire or drive. Decreased libido was a symptom that occurred at the same rate every follow-up year and was reported to be the second most frequent and distressing symptom. From a health promotion perspective, it should be considered positive that HTRs have thoughts and reflections regarding their libido. Sexual health after HTx is often discussed in terms of dysfunction and avoiding infections, although some contributions to this area have a holistic perspective [23,24]. Sexual health should be addressed and discussed from the HTRs’ perspective as it has effect on their QoL [25].

### 4.3. Gender Differences

In contrast to previous research on symptom distress [20] and chronic pain [8], there were hardly any gender differences regarding symptom distress. The significant gender difference pertaining to increased appetite presumably lacks clinical relevance as the median score was 0 in both groups, indicating no problems at all. Interestingly, it has been found that while women did not report more symptom distress than men five years post-transplant, they used negative coping styles and had problems adhering to the transplant regimen to a greater extent than men [26]. Thus, transplant professionals must be persistent and identify the unique strengths and limitations of HTRs and be gender sensitive in their assessment, thus a symptom management model would be helpful. The symptom management theory provides a generic tool for person-centred symptom management [15]. It consists of the following three domains: person, environment, and health & illness. Within these domains the person experiences symptoms and the way the symptoms affect the HTR is dependent on the three dimensions; symptom experience, components of symptom management strategies and outcomes. The best support might be provided when transplant professionals use a tool that integrates all aspects of the HTR including gender, social and cultural variations.

### 4.4. Psychological Well-Being

HTRs face many challenges. Although numerous symptoms can occur, not all of them cause distress, which is positive and should be acknowledged. Symptom occurrence is significantly higher among HTRs with poor psychological well-being and thus a target group for symptom management support. We know that HTRs deal with depression, anxiety, and other forms of psychological illness [7] along with other symptoms [3]. Our findings reveal the magnitude of symptom occurrence and distress that HTRs are supposed to manage, mainly on their own, as symptom management is a cornerstone of self-management, but self-management support is underestimated and less developed in transplant outpatient care. With poor psychological well-being the challenge inherent in symptom management as well as self-management is even bigger, raising the question of whether it is reasonable to expect HTRs to deal with it without proper support from professionals. A recent survey performed by the European Society of Organ Transplantation and its patient inclusion task force involved 352 responses from organ recipients in 27 different European countries. One of the organ recipients’ main concerns was symptoms and side-effects of the medication and they requested attention, action, and support from transplant professionals in a person-centred manner [27].

A pilot intervention including kidney recipients aimed at evaluating the feasibility of a nurse-led self-management support intervention showed that those with several medical issues still wanted to discuss other matters of a sensitive nature, such as emotional and social problems [11]. Sometimes the medical issues are in the background with no relationship between physiological functioning and self-reported health concerns [28]. A strong relationship between recovery and well-being one to five years after lung transplantation has been revealed, but no relationship between lung function and self-reported recovery [28]. This emphasizes that a more holistic approach is necessary in follow-up care.

### 4.5. Consequences and Implications

Possible overuse of immunosuppression therapy may play a role in the development of medical problems such as infections, chronic kidney disease and malignancies after HTx. A tailored immunosuppressive therapy might lower the incidence of these side-effects [29], but also tremor, pain, and diarrhoea [21,30]. A strong team approach is warranted where we move from a traditional care model that exclusively comprises medical management to chronic illness management involving emotional, role and medical management by multi-professional teams to optimize outcome [12].

In line with Ricoeur [31], we believe in the capable human being where HTRs are capable per se but face a difficult mission when it comes to symptom occurrence and symptom distress. In the pursuit of the good life, HTRs have the ability to speak, act, tell and take responsibility. Thus, we as transplant professionals are obliged to provide self-management support by means of tools and strategies to manage the symptom distress and extensive self-care demands involved in HTx, specifically medication and dealing with side-effects. It is well established that symptom burden is a barrier to returning to work, which is of concern as returning to work is an important marker of social integration and adaptation after solid organ transplantation [32]. Investigating the extent to which each individual HTR experience distressing symptoms should be considered a priority in follow-up care.

### 4.6. Methodological Considerations, Limitations, and Strengths

This study suffers from the same limitations as all cross-sectional studies. The limited information retrieved exclude an in-depth inside perspective and causality. Using self-report measures was one way of partly compensating for this lack. The exclusion of non-Swedish speaking persons systematically excludes a growing group of HTRs in Sweden and limits generalizability of the findings. There was a high turnover among the staff at the outpatient clinics at the time of data collection, which negatively affected patient recruitment, especially in the five-year follow-up group. Thus, no statistical calculations could be made in that group. A strength was the transplant specific instrument, OTSWI, which has good psychometric properties [20]. The other instrument, Psychological General Well-being (PGWB), is a generic instrument frequently used in transplant research with good internal consistency.

## 5. Conclusions

In conclusion, the most common symptoms are also the most distressing and the HTRs most burdened by symptoms are those younger than 50 years or not working or reporting poor psychological well-being. Fatigue is the strongest predictor of transplant specific well-being and explains over 60% of the variance.

This study provides a road map for transplant professionals on what to expect during each follow-up year, as symptom management is an essential part of self-management, thus enabling targeted self-management support. Screening all HTRs for symptom occurrence and distress should be mandatory, but in the case of limited resources this study also contributes information about which groups to prioritize.

## Figures and Tables

**Table 1 ijerph-17-08052-t001:** Demographics, indications for transplantation and relevant clinical aspects among 79 heart recipients.

Variable	*N* (%)
Female	25 (32)
Male	54 (68)
Indications for transplantation	
Dilated cardiomyopathy (different forms)	63 (80)
Other (e.g., hereditary conditions)	7 (9)
Congenital heart disease	4 (5)
Ischemic heart disease	4 (5)
Eisenmenger	1 (1)
Mechanical assistant device and time on ventilator	
Left ventricular assist device before Htx	24 (30)
> 48 h on ventilator after Htx	16 (20)
< 48 h on ventilator after Htx	61 (77)
Missing data regarding ventilator	2 (3)
Immunosuppressive medication and rejections	
Cyclosporin	18 (23)
Tacrolimus	59 (75)
Mycophenolic acid	72 (91)
Azathioprine	3 (4)
Steroids	20 (25)
Other drugs (for example Everolimus)	23 (29)
Persons having one or more cellular rejections	23 (29)

Htx = Heart transplantation

**Table 2 ijerph-17-08052-t002:** Symptom occurrence and symptom distress among 79 heart recipients one to five years after heart transplantation.

Symptom	Not at All (%)	A Little Bit (%)	Somewhat (%)	Quite a Bit (%)	Very Much (%)
I am breathless *	55	33	8	3	0
I need to rest because I am breathless *	68	22	8	1	0
I am bloated *	69	23	6	1	0
I feel nauseous *	82	1	3	3	0
I have oral fungus *	96	4	0	0	0
I have oral herpes *	88	12	0	0	0
I have increased appetite *	58	28	6	3	0
I have decreased appetite *	91	3	5	1	0
I have dyspepsia *	71	26	3	1	0
I am constipated *	87	12	1	0	0
I have diarrhoea **	70	22	1	5	1
My skin is itchy **	81	13	5	1	0
I have headache	57	28	10	3	1
There is a burning pain in my hands *	91	6	0	1	1
There is a numb and stabbing pain in my hands *	79	13	5	0	3
My hands are trembling *	45	32	15	4	3
I feel dizzy *	58	33	4	3	3
I feel sad **	60	22	13	4	1
My looks make me embarrassed *	74	18	5	3	0
My libido is decreased *	50	29	8	6	6

* One missing response; ** Two missing responses.

**Table 3 ijerph-17-08052-t003:** Symptom occurrence separated on follow-up among 79 heart recipients reporting some degree of symptom occurrence.

Symptom	1 Year *n* = 28 (%)	2 Years *n* = 17 (%)	3 Years *n* = 11 (%)	4 Years *n* = 17 (%)	5 Years *n* = 6 (%)
I am breathless *	59	44	27	30	50
I need to rest because I am breathless *	48	31	9	12	33
I am bloated *	50	18	27	24	17
I feel nauseous *	25	24	18	0	33
I have oral fungus *	0	6	0	12	0
I have oral herpes *	14	6	18	5	17
I have increased appetite *	52	35	45	24	67
I have decreased appetite *	15	6	9	6	0
I have dyspepsia *	42	12	27	24	50
I am constipated *	19	18	9	6	0
I have diarrhoea **	39	18	46	12	50
My skin is itchy **	31	12	9	18	17
I have headache	52	30	36	24	50
There is a burning pain in my hands *	4	18	0	6	33
There is a numb and stabbing pain in my hands *	23	30	9	12	33
My hands are trembling *	60	60	46	41	67
I feel dizzy *	41	41	64	29	50
I feel sad **	35	30	55	35	50
My looks make me embarrassed *	33	24	9	26	33
My libido is decreased *	44	59	46	47	67

* One missing response; ** Two missing responses.

**Table 4 ijerph-17-08052-t004:** The ten most frequent and distressing symptoms among 79 heart recipients one to five years after heart transplantation.

Rank Order	Most Prevalent Symptom	Most Distressing Symptom
1	My hands are trembling	My hands are trembling
2	My libido is decreased	My libido is decreased
3	I am breathless	I feel sad
4	I have increased appetite	I have headache
5	I have headache	I am breathless
6	I feel dizzy	I need to rest because I am breathless
7	I feel sad	I have increased appetite
8	I need to rest because I am breathless	I feel dizzy
9	I am bloated	I am bloated
10	I have diarrhoea	I have diarrhoea

**Table 5 ijerph-17-08052-t005:** Socio-demographic data among 79 heart recipients one to five years after heart transplantation.

Variable	*N* (%)
Marital status	
Single	15 (19)
Married/Cohabiting	51 (65)
Divorced/separated	13 (16)
Living arrangements	
Living alone	20 (25)
Single with children	3 (4)
Cohabiting without children	30 (38)
Cohabiting with children	13 (17)
Other	10 (13)
Missing	3 (3)
Level of education	
Primary	7 (9)
Second level	46 (58)
University level	26 (33)
Employment status	
Employed (full time/part time)	32 (40)
Not employed	33 (42)
Own company-working	9 (11)
Own company-not working	3 (4)
Missing data	2 (3)
Work ability	
Able to work fulltime/part time	54 (68)
Unable to work or study	20 (25)
Missing data	5 (7)
Sick leave or retired	
Temporary sick leave full time/part time	18 (23)
Permanent sick leave full time/part time	14 (18)
Retired	14 (18)

**Table 6 ijerph-17-08052-t006:** Symptom occurrence and differences between heart recipients with good psychological well-being versus poor psychological well-being.

Symptom	Good Psychological Well-Being (%)	Poor Psychological Well-Being (%)	*p*-Value
Sleep problems	81	92	0.004
Fatigue	63	89	0.007
Mood	27	59	0.002
I am worried about my financial situation	41	62	0.020
I am breathless *	30	60	0.012
I need to rest because I am breathless *	19	38	0.011
I am bloated *	19	41	0.036
I have increased appetite *	30	51	0.025
I have headache	27	57	0.003
My hands are trembling *	40	68	0.011
I feel sad **	19	61	≤0.001
My looks make me embarrassed *	5	41	≤0.001
My libido is decreased	54	41	NS

NS = Not significant; * One missing response, ** Two missing responses.

**Table 7 ijerph-17-08052-t007:** Logistic Regression Predicting Likelihood of Reporting Poor psychological well-being among 79 heart recipients one to five years after heart transplantation.

	*p*	Odds Ratio	95.0% C.I. for Odds Ratio
Lower	Upper
Age	0.046	0.964	0.929	0.999
Sex	0.352	0.584	0.188	1.813
OTSWI-Fatigue	0.003	1.430	1.128	1.814
Constant	0.226	3.833

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
