# Peer review of "Symptom Occurrence and Distress after Heart Transplantation—A Nationwide Cross-Sectional Cohort Study"

_ijerph, 2020, doi:10.3390/ijerph17218052_

Round 1

Reviewer 1 Report

As an abdominal transplant surgeon, I found your paper interesting and very helpful in understanding the issues with which our transplanted patients have to contend. I viewed it more as an observational study as so many data points were not collected and there were a large group of your patients you did not even include, that makes it harder to be definitive.

I have a few typographical issues:

1.) Page 2, line 53, remove the underline between "adopted" and "within".

2.) Page 4, lines 149 and 150: I am unclear what is meant by the term (Dnr: blinded). If this is a term that is clear to the majority and I am the minority, so be it, but I do not understand what is meant. I assume it has something to do with the type of study it is.

3.) Page 5, lines 160-163: I did not understand from where this data came that is summarized in the paragraph as there is nothing about sleep problems, fatigue, joint and muscle pain, etc, in Table 2.

4.) Page 7, line 177: The reported p value does not make sense. Did you mean (p=.001)? Also I believe you meant to write "nauseous" and not "nausea".

5.) Page 7, line 195: Please make problem plural.

6.) Page 9, line 245: Please remove the comma after "hands".

Author Response

Dear reviewer.

We deeply appreciate Your comments on our manuscript. Below You will find our changes of the manuscript according to Your suggestions.

Reviewer comment= R.C.

Author response = A.R

R.C:  I viewed it more as an observational study as so many data points were not collected and there were a large group of your patients you did not even include, that makes it harder to be definitive.

A.R. We agree that the study could be viewed as observational.

R.C.: Page 2, line 53, remove the underline between "adopted" and "within".

A.R.: Thank you for noticing. It has been done.

R.C.: Page 4, lines 149 and 150: I am unclear what is meant by the term (Dnr: blinded). If this is a term that is clear to the majority and I am the minority, so be it, but I do not understand what is meant. I assume it has something to do with the type of study it is.

A.R.: We apologize for the confusion. The ethic approval code was blinded but is now inserted in the manuscript.

R.C.:  Page 5, lines 160-163: I did not understand from where this data came that is summarized in the paragraph as there is nothing about sleep problems, fatigue, joint and muscle pain, etc, in Table 2.

A.R.: Thank you for this remark that enables us to clarify. The data stem from the part of the OTSWI-questionnaire that summarize 20 questions into the 8 factors (fatigue, joint and muscle pain, cognitive functioning, basic activities of daily life, sleep problems, mood, foot pain, and financial situation) to explore the transplant specific well-being.

The data shown in table 2 stem from the part of OTSWI-questionnaire that measures symptom distress among 20 transplant specific symptoms.

You will find a clarification in the text starting at line 161. We hope that it will make it more clear to the reader.

R.C.: Page 7, line 177: The reported p value does not make sense. Did you mean (p=.001)? Also I believe you meant to write "nauseous" and not "nausea".

A.R.: Yes, we meant (p=.001). Thank you for noticing. we also changed the word "nausea" to "nauseous".

R.C.: Page 7, line 195: Please make problem plural. Page 9, line 245: Please remove the comma after "hands"

A.R.: Thank you for noticing. It has been changed.

Once again, we deeply appreciate Your efforts to improve our manuscript. 

Reviewer 2 Report

The authors of the submitted manuscript describe their research findings concerning the frequency of symptoms in patients having received heart tpx. These results seem interesting as they fill in the knowledge  of symptoms management for this group of patients and their caretakers. Althouhg this is an interesting study there are some issues that have to be taken care of.

-Abstract should be remodified as a conclusive sentence lacks from the end of it. The reader cannot comprehend the scope of the manuscript. 

-Even there were no need for controlling the influence of gender differences in the dimensions assessed, differences in the outcome depending on the genders should be mentioned/highlighted as there is a substantial difference between the numbers of each gender cohort. 

-Have you applied multiple comparisons testing parameters for validating your analysis?

-Why were not included non-swedish speaking patients?

-I think that the following studies should be included and further discussed since their context lies in the same scope of the manuscript (Chou YY et al, The Journal of Cardiovascular Nursing, 2017 and Rosenberger EM et al, Curr Opin Organ Transplant, 2012)

Author Response

Dear reviewer.

We appreciate Your time and effort to improve our manuscript.

Below you will find our response according to Your comments.

Reviewer comment= R.C.

Author response = A.R

R.C.: -Abstract should be remodified as a conclusive sentence lacks from the end of it. The reader cannot comprehend the scope of the manuscript. 

A.R.: We appreciate this comment and agree that it is missing.  We have now added a conclusive sentence.  

R.C.:Even there were no need for controlling the influence of gender differences in the dimensions assessed, differences in the outcome depending on the genders should be mentioned/highlighted as there is a substantial difference between the numbers of each gender cohort.

A.R.: There is a great difference between the numbers of each gender in this study. Though it reflects the population of heart recipients in Sweden with approximately 70 % male heart recipients and 30% female. Because of that  we didn´t mention this further in the study. This might not be known to the reader, so we have made some changes in the text to explain the proportions of the heart recipient population in Sweden at line 101. 

R.C.: Have you applied multiple comparisons testing parameters for validating your analysis?

A.R.: No, we have not. 

R.C.: -Why were not included non-swedish speaking patients?

A.R.: We excluded non-swedish speaking patients due to the fact that we didn't use questionnaires translated to multiple languages. That is of course a weakness of the study since Sweden is a multi-ethnic country. 

R.C.: -I think that the following studies should be included and further discussed since their context lies in the same scope of the manuscript (Chou YY et al, The Journal of Cardiovascular Nursing, 2017 and Rosenberger EM et al, Curr Opin Organ Transplant, 2012

A.R.: We deeply appreciate your suggestions on further references to support the manuscript.

In the reference list, no 7 we refer to Dew et al. 2015. Depression and anxiety as risk factors for morbidity and mortality after organ transplantation: A systematic review and meta-analysis. Transplantation 2015, 100, 988 – 1003. doi: 10.1097/TP.0000000000000901. This review includes the suggested paper by Rosenberger et a., 2012 and thus a part of our reference list already. 

Regarding Chou et al., we appreciate the suggestion and is now  discussing it in relation to our findings.

We appreciate your efforts to improve our manuscript. 

Reviewer 3 Report

The authors investigated the symptoms and distress in patients who underwent heart transplantation. The present manuscript seems to be well-written, however, there are several problems to be solved.                                                                                                                                                              #1 Data collection and inclusion criteria seem not to be clear. The authors should provide the flow-chart of data collection in figure. If patients had been followed at other institution, why were not they invited to the present study? In the "Method" section, 90 patients were included in the study, however, data were shown in 79 patients, such as in Table 1. Which number was right?                                                                                                                                            #2 The authors should provide any information regarding the contents and numbers of medications in Table 1. Especially, the authors should show the relationship between symptom and taking beta-receptor blockers. In addition, were there any patients who had taken antidepressant?                                                                                                                                                            #3 Alike #2, the authors should provide the relationship between the symptoms and the status of heart failure, such as NYHA classification or BNP levels.                                                                                                                                  #4 In some tables, the factors may be more favorable, in the aligned left than in the aligned on the center.                

Author Response

Dear reviewer.

We would like to thank You for Your effort improving this manuscript. Below you will find my responses according to your comments.

Reviewer comment= R.C.

Author response = A.R

R.C.: 1 Data collection and inclusion criteria seem not to be clear. The authors should provide the flow-chart of data collection in figure. If patients had been followed at other institution, why were not they invited to the present study? In the "Method" section, 90 patients were included in the study, however, data were shown in 79 patients, such as in Table 1. Which number was right? 

A.R.: We agree with you that a flow-chart would have been the ideal way of presenting the selection of the participants. However, this is impossible as described in the manuscript. The reasons was unexpeted and heavy staff turnover in the out patient clinics involved during data collection. The sample was collected consecutively but should be viewed also as a convenient sample due to the circumstances mentioned. There were not enough logistic resources to enable inclusion from all out patient clinics dealing with heart recipients in our country. 

Initially 90 persons were included but 11 patients didn't return the questionnaires and constituted the external drop out.

R.C.; The authors should provide any information regarding the contents and numbers of medications in Table 1. Especially, the authors should show the relationship between symptom and taking beta-receptor blockers. In addition, were there any patients who had taken antidepressant?     

A.R.; We agree that it migh have been useful with a more pharmacological approach to the research question. However, we didn't record numbers of medications the way You suggest. Further, all heart transplant recipients are treated with beta blockers and thus making it difficult to outline any specific relationships with the invesitgated symptoms. There were probably some participants on SSRI- medication. However, we were not allowed according to the research ethical approval to collect data about that specific medication. 

R.C.: 3 Alike #2, the authors should provide the relationship between the symptoms and the status of heart failure, such as NYHA classification or BNP levels.  

A.R.: According to our inclusion criteria heart recipients with CHF due to extensive graft dysfunction or graft rejection would not be included. The patients included were heart recipients with good graft function and thus not classified according to NYHA- criteria for CHF. Since the focus of the study was symptoms experienced from an illness perspective we didn't explore any biomedical signs. 

R.C.:  In some tables, the factors may be more favorable, in the aligned left than in the aligned on the center.                

A.R.: We agree. However, the tables are set according to the author guidelines and preparation of the manuscript has been following the dashboard. 

We would like to take the opportunity to thank You once again for your important comments that have inspired us regarding possible data analysis and research questions in future studies.  

Round 2

Reviewer 2 Report

No further comments. I appreciates the authors' honesty and effort to reply to the comments as much as possible.